# High-intensity Interval Training Promotes the Shift to a Health-Supporting Dietary Pattern in Young Adults

**DOI:** 10.3390/nu12030843

**Published:** 2020-03-21

**Authors:** Sabrina Donati Zeppa, Davide Sisti, Stefano Amatori, Marco Gervasi, Deborah Agostini, Giovanni Piccoli, Alexander Bertuccioli, Marco B.L. Rocchi, Vilberto Stocchi, Piero Sestili

**Affiliations:** Department of Biomolecular Sciences, University of Urbino Carlo Bo, 61029 (PU) Urbino, Italy; davide.sisti@uniurb.it (D.S.); s.amatori1@campus.uniurb.it (S.A.); deborah.agostini@uniurb.it (D.A.); giovanni.piccoli@uniurb.it (G.P.); alexander.bertuccioli@uniurb.it (A.B.); vilberto.stocchi@uniurb.it (V.S.); piero.sestili@uniurb.it (P.S.)

**Keywords:** physical exercise, eating behaviour, diet regulation, healthy lifestyle

## Abstract

A healthy lifestyle is based on a correct diet and regular exercise. Little is known about the effect of different types of exercise on dietary preferences. To address the question of whether high-intensity interval training (HIIT) could modulate spontaneous food choices, an experimental study was carried out on 32 young, healthy normal-weight subjects. The spontaneous diet of each subject has been monitored over nine weeks of indoor-cycling training, divided into three mesocycles with an incremental pattern: total energy intake, macronutrients and micronutrients have been analysed. A two-way mixed model has been used to assess differences in dietary variables; a principal factor analysis has been performed to identify sample subgroups. An increased energy intake (+17.8% at T3; *p* < 0.01) has been observed, although macronutrients’ proportions did not vary over time, without differences between sexes. An increase of free fat mass was found in the last mesocycle (+3.8%), without an augmentation of body weight, when, despite the increased training load, a stabilization of energy intake occurred. Three different subgroups characterized by different dietary modifications could be identified among participants that showed a common trend towards a healthier diet. Nine weeks of HIIT promoted a spontaneous modulation of food choices and regulation of dietary intake in young normal-weight subjects aged 21–24. Importantly, this life-period is critical to lay the foundation of correct lifestyles to prevent metabolic diseases and secure a healthy future with advancing age.

## 1. Introduction

Physical exercise and nutrition (diet) are the two pillars of wellness on which correct lifestyle guidelines are based, although it is undeniable that only a small proportion of the population in industrialized countries follows an active lifestyle. Physical inactivity and an incorrect diet are involved in the aetiology of obesity. As hypothesized and subsequently verified at a preliminary level in the mouse model by Koch et al. [1], the change in energy transfer capacity that occurs through exercise capacity can be considered the central mechanistic determinant in the division between disease and health, where the continuous cellular and molecular reorganization characteristic of living organisms constitutes a dynamic evolutionary strategy providing a selective advantage to subjects capable of developing and maintaining a better health condition. 

Regular exercise is currently adopted along with diet and eating habits modification to counteract obesity, even if further studies are needed to clarify if they work synergically but remain separate strategies, or if they track together and share the same neurobiological substrates and pathways. The wellness effects of physical exercise are already well known, improving the control of appetite-regulating hunger-satiety and psychological mechanisms. The appetite and hunger are suppressed following exercise, in particular in the immediate post-exercise state [2]; this effect, known as “exercise-induced anorexia”, is linked to the suppression of orexigenic hormones, such as ghrelin, and the increase of satiety hormones, such as peptide YY [3]. It is known that regular exercise positively influences mood and self-control in several areas and effectively counteracts binge eating also in response to negative emotions [4,5]. Mechanisms involved in hunger–satiety are better regulated in active people in whom eating in response to negative emotions is less frequent. Furthermore, even if energy request during physical exercise can lead some subjects to augment caloric intake, an implementation in exercise causes a negative energy balance, also due to improved homeostatic control of appetite [6]. Several studies suggest that regular physical activity can positively affect eating behaviour and food choices in a healthy direction, helping in correcting inappropriate long-term eating habits [7].

This relationship between physical exercise and diet opens the possibility of considering exercise as a gateway behaviour that could lead to dietary behaviour changes [8]. To the authors’ knowledge, the available studies investigating the relationship between exercise and dietary behaviours are characterized by short duration protocols (days or weeks), and studies of longer duration (several months) are lacking [9]. Furthermore, the influence of exercise on dietary preferences is dependent on the type, method, frequency, duration and intensity of exercise performed; the sample sizes and characteristics may also affect the results [9]. Recently, Joo et al. [10] reported that aerobic exercise of longer duration or higher intensity led to different dietary patterns, even though the trend was toward healthier diets in any case. 

Traditionally, physical activity to improve health in the non-athlete population has been characterized by constant low- to moderate-intensity exercise. However, in the last decade, growing evidence proposed that vigorous exercise may produce more benefits than moderate-intensity exercise alone; in this context, high-intensity interval training (HIIT) is considered one of the most effective activities to improve cardiorespiratory and metabolic function and maximize health outcomes [11]. Buchheit and Laursen [12] defined HIIT as “repeated short-to-long bouts of a rather high-intensity exercise interspersed with recovery periods”. The exercise intensity during a HIIT session can be individualized, such that two subjects may train at different speeds or power (external load), but with the same relative intensity, as %VO_2_peak or %HRmax (internal load). This implies that HIIT is an exercise mode that could be applied and finely tuned to different types of population [11]. HIIT can be manipulated by several different parameters, such as duration and intensity of work phases, number of series and repetitions, work to rest ratios, etc.; indeed, distinct protocols can differently affect the adaptive responses to the exercise [13]. 

In this study, we investigated whether a HIIT program could play a role in modulating spontaneous food choices and how such food choices were more or less healthy. The aim was to evaluate if a specific incremental indoor cycling HIIT program was able, in itself, to modify eating habits without dietary counselling. This might pave the way to the development and implementation of effective intervention programs to maintain lifelong health and prevent metabolic pathologies. The experimental study was carried out on young subjects in health and of normal weight.

## 2. Materials and Methods 

### 2.1. Study Design

This was a within-subject training study. Dietary habits of a sample of young, healthy subjects were evaluated before, during and after a period of high-intensity interval training. 

### 2.2. Participants

Thirty-two healthy collegiate subjects (20 males: age 22.6 ± 1.7 years, height 175.5 ± 6.5 cm, weight 67.9 ± 9.9 kg, BMI 22.0 ± 2.7 kg/m^2^; 12 females: age 21.5 ± 0.8 years, height 159.5 ± 4.7 cm, weight 52.8 ± 5.2 kg, BMI 20.6 ± 1.2 kg/m^2^) were recruited. The exclusion criteria were major cardiovascular disease risks, musculoskeletal injuries, upper respiratory infections, smoking and consumption of any medicine in the past three months. With respect to nutritional habits, exclusion criteria were lactose intolerance, celiac disease, food allergies and particular dietary regimens (vegetarian, vegan, ketogenic, Palaeolithic, intermittent fasting or less common diets). 

The subjects were asked to answer a set of questions, taken from Malek et al. [14], regarding the mode, frequency, duration, length of time performing habitual physical activity and intensity of the exercise performed. All participants performed not more than one 60 min leisure walking or jogging session per week, in the three months before the start of the study. The maximal oxygen consumption (VO_2_max) of the participants was directly measured with an incremental ramp test (as described below); VO_2_max values were 42.0 ± 5.9 mL/kg/min for men and 32.2 ± 6.5 mL/kg/min for women, in line with normative values of non-active young people already reported in the literature [15,16].

The participants were advised to maintain their dietary routine and to abstain from using dietary supplements, nutraceuticals or drugs, over the study period. The participants were also instructed to refrain from all training activities except the sessions included in the experimental design. Following a medical health-screening, all participants provided written informed consent to participate in the study, which was approved by the Ethical Committee of the University of Urbino Carlo Bo, Italy (02/2017, date of approval July 10, 2017) and was conducted in agreement with the Declaration of Helsinki for research with human volunteers (1975).

### 2.3. Training Protocol

Thirty-six indoor cycling training sessions over nine weeks were performed (see Figure 1). The training program was structured in three mesocycles, with an incremental pattern: both frequency and duration of the sessions were increased. Before the beginning of the training period, an incremental ramp test was performed by every subject, in order to set the individual training zones, based on the heart rate (HR) recorded at the first and second lactate thresholds (LTs); maximal oxygen consumption (VO_2_max) was also assessed. For further information about training zones definition, see Gervasi et al. [17]. Each session was designed following the same training intensity distribution, based on a polarized model, with around 70% of session time spent in zone 1, 10% spent in zone 2 and 20% spent in zone 3, according to Seiler and Kjerland [18]. Each session was composed of a warm-up, a high-intensity interval exercise, and a cool down, as widely employed in the indoor cycling community. Intervals were performed at different intensities and durations, in order to match the polarized training intensity distribution (an example of a workout is presented in Figure 2). As a general guideline, intervals longer than 4 min were performed in zone 2, while intervals between 2 and 4 min were performed in zone 3. Recovery periods between intervals were performed in zone 1. During the training sessions, HR of each subject was monitored, and the values were projected onto the wall, as a percentage of maximal HR; the subjects were asked to maintain the same intensity zone of the instructor, based on their HR zones. All participants completed the whole training program; three subjects had to quit a training session before the end due to personal issues, but this was not considered a problem for the statistical analyses.

### 2.4. Body Composition Assessment

Weight and BMI (Body Mass Index) were measured for each participant. Body composition was assessed through electrical bioimpedance (Arkray BIA101 Sport Edition), at T0 before the beginning of the training, and subsequently after each mesocycle of training (T1, T2, T3). The measurements were taken in the morning nearest to the last session, after six or more hours of fasting. Percentage of Free-Fat Mass (FFM) was reported for each mesocycle.

### 2.5. Diet Monitoring

Diet monitoring of subjects started two weeks before the start of training (T0) until the end of the training protocol (T3) and was performed by daily call interviews after dinner. In order to choose the most representative portions according to the indications of the subjects, a PHOTOdietometer was provided to each of them. The PHOTOdietometer is a visual scale which, by observing the different images, helps subjects to identify and objectively refer the portion of food taken to volume, weight in grams and/or carbohydrate content. Information about mealtime and its characteristics in terms of cooking, seasonings and quantities were also provided. Data were collected and processed using MètaDieta software (METEDA Srl, San Benedetto del Tronto, Italy), and the following variables were considered for elaborations:total energy intake and macronutrient quantity;starch, soluble and insoluble fibre, glycaemic index and glycaemic load;saturated, monounsaturated, polyunsaturated fat, omega-3, Omega-6, cholesterol;iron, vitamin C, A, E.

### 2.6. Statistical Analysis

Descriptive statistics were performed using mean and standard deviation, where appropriate. Percentage variations were also reported in diet habits for each macro- or micronutrient considered. In order to verify the weight, free-fat mass, macro- and micronutrient mesocycle-dependent changes, two ways mixed design (MANOVA for repeated measures) has been performed. Time was within-subjects 4 levels factor (T0, T1, T2, T3). T0 mean values were revealed during 2 weeks before starting training, T1 relative to the first mesocycle, T2 to the second mesocycle and finally T3 to the third. Sex was two levels between groups factor. Contrasts are used to test for differences among the levels of a between-subjects factor; simple contrast compares the mean of each level (T1, T2, T3) to the mean of T0 level. Overall and partial Eta squared were used as effect size estimation. Mauchly’s Test of Sphericity was performed; when epsilon was >0.75, the Huynh-Feldt correction was applied, and when epsilon was <0.75, the Greenhouse-Geisser correction was applied [19]. Wilks’ lambda was considered as an appropriate multivariate test statistic. A Principal Axis Factor with a Varimax (orthogonal) rotation of delta (T3-T0) of 16 nutritional variables above considered was conducted on data gathered from participants. Kaiser-Meyer Olkin (KMO) measure of sampling adequacy was also performed; the minimum acceptable value for KMO is 0.6, although the ideal is over 0.70. Only factors with an eigenvalue of ≥1 were considered. The variance percentage accounted for by each component to the total variance was also reported. All elaborations were conducted with α = 0.05. Elaborations and graphics were obtained using Prism 6 (GraphPad, La Jolla, California) and SPSS version 20.0 (SPSS Inc., Chicago, IL, USA).

## 3. Results

### 3.1. Weight, Body Composition, Macro- and Micronutrients

Mauchly’s Test of Sphericity indicated that the assumption of sphericity had been violated for all dependent variables, and therefore a Greenhouse-Geisser correction was used. The results showed no gender difference (time × sex interaction, Wilks’ lambda = 0.63; *F*(57,215) = 0.634, *p* = 0.98); this means that the effect of physical exercise over the three mesocycles is not different between sexes. A significant effect of time on all dependent variables (Wilks’ lambda = 0.153; *F*(57,215) = 3.314, *p* < 0.001) was observed; this finding suggests that physical exercise led subjects to modify their diet. However, this modification had no effect on body weight that did not change over time, either in males or females but, interestingly, the percentage of free-fat mass significantly increased at T3 (+3.8%, *p* < 0.001), without differences between sexes (*p* = 0.097) (Figure 3; Table 1).

Univariate tests showed significant association with time for the following nutrients: total energy intake (*p* < 0.001), protein (*p* = 0.001), carbohydrate (*p* = 0.03), starch (*p* < 0.001), glycemic index (*p* = 0.02), glycemic load (*p* < 0.001), total fat (*p* = 0.02), saturated (*p* = 0.009) fat, monounsaturated (*p* = 0.017) fat, polyunsaturated (*p* = 0.009) fat, cholesterol (*p* = 0.004), iron (*p* = 0.028), and vitamin C (*p* = 0.032) (Figure 4 and Figure 5). 

For each of the above significant variables, considering simple contrast analysis among T0 values and subsequent mesocycles (T1, T2 and T3), it was possible to highlight the following results, which are depicted in Figure 5: 

(I) total energy intake increased significantly at the T1 mesocycle (+9.4%, *p* = 0.001), and raised a +17.8% value at T3 (*p* < 0.01); ii) negligible differences were found in the mean proportions of energy deriving from macronutrients intake in the different mesocycles: carbohydrate, fat and protein accounted for 49.6%, 33.1% and 17.6% over the entire experimental protocol, respectively; 

(II) the net intake of the macronutrients showed an overall increase in the course of the mesocycles. In particular protein intake increased at T1 (+12.9%, *p* = 0.003) and reached a +18.5% value at T3 (*p* < 0.01); fats and carbohydrates showed a similar trend with no significant increase at T1, +21.5% (*p* < 0.01) and +19.2% (*p* < 0.01) at T2, +19.4% (*p* = 0.015) and +17.1% (*p* = 0.036) at T3, respectively; 

(III) a more detailed analysis of lipid intake composition showed that saturated (T1: +12.4%, *p* < 0.01; T2: +22.0%, *p* < 0.01; T3: +21.4%, *p* = 0.012) and monounsaturated fats (T1: +9.1%, *p* < 0.01; T2: +15.8%, *p* = 0.012; T3: +14.7%, *p* = 0.023) increased in all the mesocycles, whilst polyunsaturated ones showed a significant increase only at T2 (+21.9%; *p* = 0.016) and T3 (+27.4%, *p* = 0.014). Although non-significant gender differences have been found with regard to total fat intake (*p* = 0.25), females showed a slight positive trend in saturated fat consumption (*p* = 0.07). Cholesterol showed a decrease at T1 (−11.1%, *p* < 0.001), starch and glycemic load increased at T2 (+22.4%, *p* < 0.01; +20.5%, *p* < 0.001), and at T3 (+30.3%, *p* = 0.001; +25.8%, *p* < 0.001), respectively. Glycemic index decreased significantly in T3 (−2.6%, *p* = 0.028). Finally iron (+17.7%, *p* = 0.037) increased at T3.

### 3.2. Factor Analysis

The results of an orthogonal rotation (Varimax rotation) of the solution are shown in Table 2. The analysis yielded a three-factor solution; the Kaiser-Meyer Olkin measure of sampling adequacy was 0.785, above the minimum ideal value. Seven items loaded in Factor 1: these are all related to an increase in energy intake, due to augmented consumption of macronutrients (carbohydrates, proteins and fats) and Vitamin E. Three items loaded in Factor 2: increased intake of Omega-3, Omega-6 and polyunsaturated fats. Finally, six items loaded in Factor 3, showing an increased intake of fibres, Vitamin A and C, starch and iron.

## 4. Discussion

In this study, the influence of exercise training on habitual dietary patterns of young sedentary adults has been evaluated. Despite increased awareness, public health campaigns and novel therapies, worldwide obesity in adults has greatly increased in the last ~40 years [20].

Lifestyle-based interventions focusing on diet and physical activity can produce a 7–10% drop in initial body weight and represent the cornerstone of weight-loss approaches [21]; however this condition is not maintained over time, with frequent weight gain relapses known as “yo-yo effect” [22]. A better understanding of the connection between the two integrated approaches, physical activity and eating behaviours, is necessary to prevent and limit the failure of long-term interventions. Notably, to the authors’ knowledge, little is known about how changes in one of these behaviours can support the long-term maintenance of the change in the other.

Physical exercise contributes to a negative energy balance by increasing energy expenditure, but the role of exercise on food intake is not yet fully understood. Changes in exercise behaviour can also influence nutrition habits by application of self-regulatory psychological resources across behaviours; in particular exercise can facilitate improved fruit/vegetable consumption in young adulthood, a life-period critical to lay the foundation, based on the adoption of a health behaviour, for the prevention of metabolic diseases and for securing a healthy future with advancing age [23]. Joo et al. [10] suggested that food choices differ depending on the type (intensity and duration) of the exercise performed. Despite this, to the authors’ knowledge, scientific literature regarding the effects of different exercise modalities on dietary habits is still scarce.

HIIT has been proposed as an effective training modality to promote fat mass reduction while preserving free-fat mass, by increasing energy expenditure and improving appetite control [24,25]. This potential anorexigenic effect does not seem to occur with moderate-intensity continuous training (MICT) [25]. HIIT has been reported to improve endurance performance to a greater extent than MICT, both in sedentary and recreationally active individuals. This may be due to an up-regulated contribution of both aerobic and anaerobic metabolism to the energy demand, a consequently enhanced availability of ATP and improved energy status of working muscles [26]. Moreover, interval exercise has been shown to produce a higher EPOC (excess post-exercise oxygen consumption) in respect to moderate-intensity exercise; this is proposed to be one of the factors that induce higher weight-loss after interval exercise [27]. Importantly, different HIIT protocols may produce different adaptive responses to the exercise: training parameters such as duration and intensity of work and recovery phases can be manipulated to conveniently and precisely adapt HIIT to different sub-populations varying in age, gender, BMI and healthiness features [11].

### 4.1. Body Composition 

No changes in weight have been observed throughout the study; on the contrary, the free-fat mass showed an increase in the last mesocycle. The first six weeks of exercise, indeed, did not produce any change in body composition, but in the last three weeks—i.e., when the frequency of training augmented up to five sessions per week—an increase in muscle mass has been found. The increase in training load in the third mesocycle is not accompanied by a higher energy intake in comparison to the previous weeks, as happened comparing the first mesocycle with baseline and the second mesocycle with the first. As previously reported by Scheurink et al. [28], the exercise-induced increased energy expenditure is followed by a compensatory increase over the long term; despite this, the not-augmented energy intake in the third mesocycle reported herein supports the hypothesis that a metabolic adaptation may occur. Furthermore, it is also possible that the influence of these factors may differ between sedentary subjects (which were enrolled in our study) and athletes by virtue of their particular physiological and psychological adaptations that include an improved ability to regulate energy balance and/or increased sensitivity to satiety signals [29]. 

The reduction of fat mass shown after the third mesocycle is in accordance with previous studies [30,31] that found a decrease in body fat mass only with higher training frequency (3 times/week vs. 1 or 2 times/week). It was concluded that the training frequency is associated with fat mass loss. Accordingly, an increase in free-fat mass after a HIIT period was reported by other authors [13,32], even though a limitation of these two studies was that they focused on overweight and obese women. Finally and along the same lines, HIIT has been demonstrated to be efficient in improving body composition in normal-weight subjects [33,34].

### 4.2. Energy and Macronutrients Intake

As previously discussed, an increase in the energy intake has been reported in the first two mesocycles of training in respect to baseline control, but no further increase has been shown in mesocycle three; macronutrients’ proportions remained relatively stable over time. These results are consistent with previous data by Miguet et al. [25], who described an augmented energy intake after 16 weeks of training in obese adolescents; moreover, these authors reported that—although the overall intake (grams) of macronutrients increased following the exercise intervention—the proportions remained approximately equal to pre-training. A slight increase in fat consumption was found in females [25]: notably we found a similar trend in women. High-intensity exercise has been shown to reduce spontaneous energy intake during lunch and dinner following the exercise bout in obese adolescents, with a positive effect on the 24-h energy balance [35]. This anorexigenic effect in obese adolescents has been confirmed by other authors [25,35] and this adaptation was due to peripheral (mainly gastro-peptides) and neurocognitive (neural responses to food cues) pathways [36]. Lactate has also been found to be a key suppressor of energy intake, perhaps due to the interaction of this molecule with glucose sensors in the central nervous system, suggesting that high-intensity exercises may have greater anorexigenic effect as compared to low-intensity ones [37].

However, for the sake of completeness, Hazell et al. [32] did not find any change in overall energy nor macronutrients intake, after six weeks of sprint interval training in respect to pre-training. Furthermore, studies [38] focusing on the effect of physical exercise on appetite did not reach a definitive consensus: indeed, inconsistent results have been obtained regarding the effect of physical exercise in altering macronutrients and micronutrients intake. In this regard, it could be speculated that a major limitation of these studies contributing to the inconsistency of the results may reside in self-reported food diaries and questionnaires on food frequencies, together with short time data collection. Indeed, self-reported food diaries are susceptible to underreporting from participants, while food frequency questionnaires present a lower underreporting risk; moreover, both these approaches are aimed to evaluate dietary intake of nutrients and foods over a given period, but are not reliable for the daily quantification of energy intake [39]. The 24-h recall shows underreporting levels of about 11% [40], but using the same method for multiple days has been shown to reduce underreporting to 4%, after 3 days [41]. Furthermore, by comparison of total energy expenditure by doubly labelled water, 24-h recalls result less underreporting than food frequency questionnaires [42]. In conclusion, the multiple-pass 24-h recall method we used in a period longer than 3 months is likely to be the best strategy. Finally, it should be considered that any strategy used to record food intake may lead participants to modify their eating habits [39]. 

### 4.3. Factor Analysis

An augmented intake of all macronutrients characterised Factor 1: this led to an absolute increase in energy intake. This factor could be summarised as “eat more”. Factor 2 was defined by increased consumption of polyunsaturated fatty acids, especially omega-3 and -6: a greater intake of these nutrients could be explained by incorporating more fish, vegetable oils, nuts and seeds into the diet. Partially overlapping results were found in 2013 by Hiza et al. [43] who, analysing a one-day diet of a sample of 8272 subjects, reported greater consumption of oils of vegetable origin together with lower consumption of saturated fatty acids in females, compared to male subjects. Finally, Factor 3 was marked by higher consumption of fibres, vitamin C and A, starch and iron: these changes could be due to an augmented consumption of fresh fruit and vegetables and cereals. Johnson et al. [44] reported a common pathway between physical activity, fruit and vegetable consumption and mental well-being. Five or more servings of fruit and vegetables may be sufficient to reduce oxidative stress and provide other health benefits, without impairing training adaptations [45]. Bellisle [2] highlighted that active persons eat more and ingest more fruits and vegetables than less active ones, even though it is not known whether these food choices are driven by biological needs (e.g., replacement of glycogen) or elicited by social and psychological factors. Furthermore, maintaining a positive iron balance is essential to avoid the effects of iron deficiency and anaemia, and to support performance in athletes [46]. 

In order to determine and prescribe the best protocols to reach a healthy weight, better knowledge of the complex interaction between physical activity and eating behaviours is needed [47]. Behavioural changes occur when an external imposition, aimed at achieving a goal, is internalized by the subject. These changes are due to brain plastic processes and modification in neurobiological substrates that provoke a neurocognitive remodelling. Physical activity and healthy choices can likely share the same pathways involving “top-down” inhibitory control, contributing to neurocognitive processes needed for effective behaviour changes [9]. Andrade et al. [6] demonstrated that women who received 12-months of behavioural intervention, including moderate and vigorous exercise, had reduced emotional overeating through eating self-regulation, suggesting that an active lifestyle can contribute to long-term weight management. Physical exercise can alleviate stress-induced unhealthy food choices [48]. Physical activity and dietary habits are likely interconnected and together could represent good strategies for sustainable weight management.

This study was subjected to some limitations. The sample size was small; however, to the authors’ knowledge, this is the only study that daily monitored participants’ diets during a structured training period. Another limitation of the present study might be that we did not specifically consider and monitor—as possible factors indicative of a “healthier dietary habit”—the presence of nutraceuticals in the diet. Indeed, although lacking an internationally accepted definition and official classification, nutraceuticals (i.e., a group of products that are more than food but less than pharmaceuticals) are increasingly considered as important constituents of a healthy diet [49]. The amount of nutraceuticals in the diet is mostly raised through the consumption of enriched foods or dietary supplements: here we asked our participants to abstain from taking dietary supplements and/or fortified foods to avoid a factor capable of influencing their instinctive diet preferences and complicating the analysis of food constituents. Hence, we deliberately excluded today’s most common sources of nutraceuticals from our investigation. However, factor analysis of our data showed an increased consumption of omega-3 and -6 fatty acids, and fibres, which are often referred to as “nutraceuticals“ [50]: in this light we can infer that their consumption tends to increase as an implicit consequence of a healthier food choice. Along these lines, it would be interesting to see whether, in the absence of the specific prescription adopted herein, HIIT leads participants to increase dietary supplements/fortified foods consumption. 

## 5. Conclusions

Physical activity and eating patterns are not independent and are closely related. Our results demonstrated that a nine weeks HIIT, progressively incremented in frequency and duration of the sessions, is capable of promoting a spontaneous and unaware modulation of food choices in young subjects, leading to a healthier diet. The factor analysis of macro- and micronutrients evidenced three subgroups with different diet habit patterns: one that “eats more”, a second that eats more fish, vegetable oils, nuts and seeds and a third that eats more fresh fruit and vegetables and cereals. Although different subgroups have been identified, the overall trend goes towards a preference for healthier food choices. Behavioural changes involving physical activity and more healthy food choices are conditions sine qua non to treat and counteract obesity and related diseases.

## Figures and Tables

**Figure 1 nutrients-12-00843-f001:**
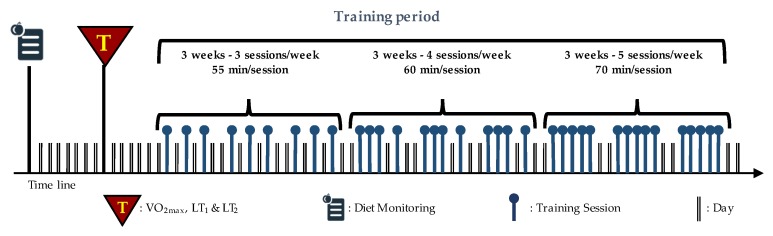
Structure of training period: nine weeks divided into three mesocycles. The frequency and duration of the sessions are also indicated. Modified with permission from Gervasi et al. [17].

**Figure 2 nutrients-12-00843-f002:**
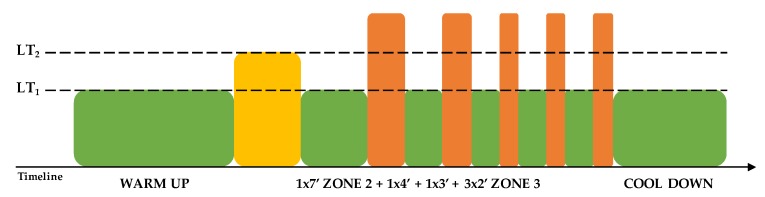
Example of a 70 min training session. LT: lactate threshold.

**Figure 3 nutrients-12-00843-f003:**
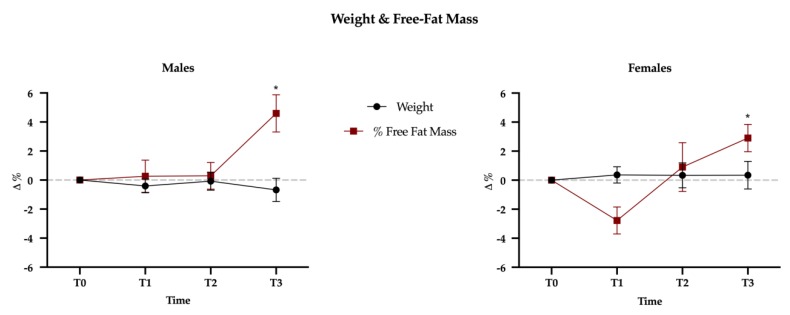
Weight and free-fat mass (FFM) variations between baseline (T0) and the three mesocycles (T1, T2, T3). Data are presented as percentage variations with standard errors.

**Figure 4 nutrients-12-00843-f004:**
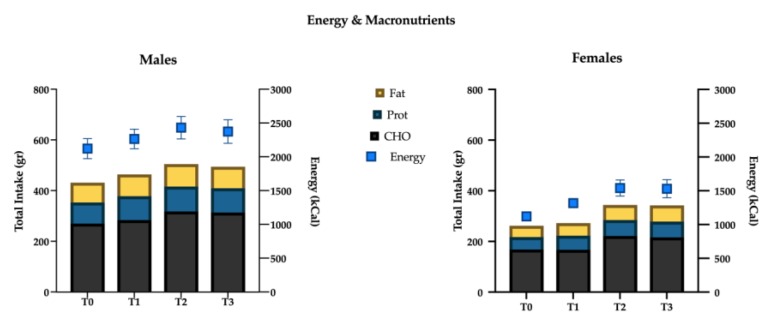
Energy intake (kcal) and macronutrients (g) variations over time.

**Figure 5 nutrients-12-00843-f005:**
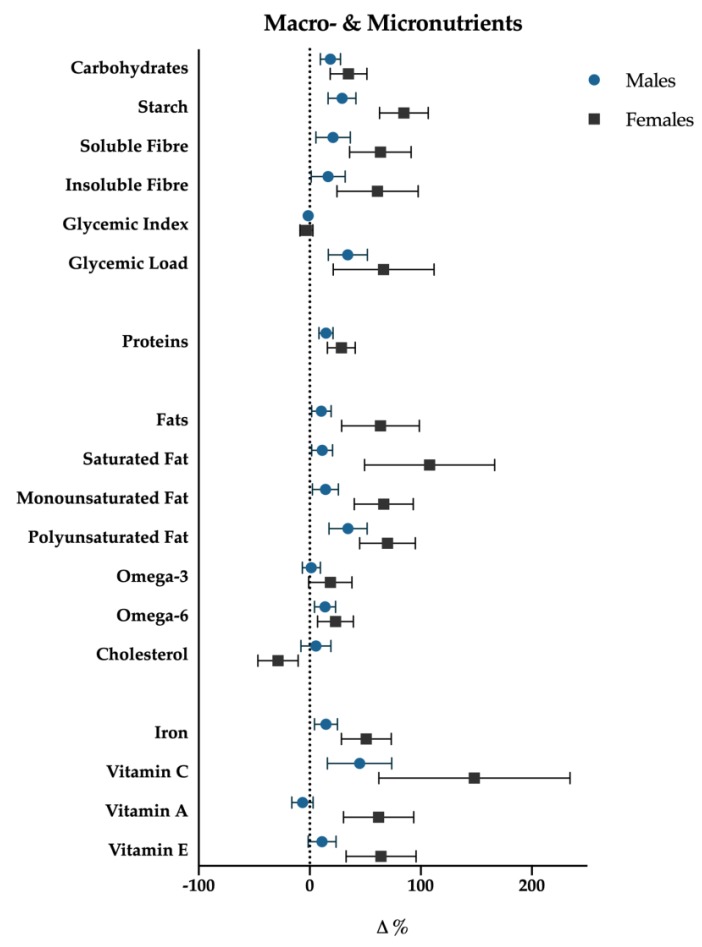
Macro- and micronutrients percentage variations (T3-T0); data are presented with standard error bars. Males are reported with blue dots, females with black squares.

**Table 1 nutrients-12-00843-t001:** Variations between baseline (T0) and end of each mesocycle (T1, T2, T3) of body mass and FFM, macro- and micronutrients; results are split between males and females and presented as mean ± standard deviations. Data highlighted in bold are statistically significant (simple contrast analysis).

	T0		T1		T2		T3	
	Males	Females	Males	Females	Males	Females	Males	Females
Weight (kg)	68.2 ± 10.0	52.9 ± 5.4	67.8 ± 9.3	53.0 ± 5.3	67.9 ± 8.6	53.0 ± 5.6	67.4 ± 7.9	53.0 ± 5.5
Free-Fat Mass (%)	83.7 ± 5.9	75.0 ± 2.9	83.7 ± 5.9	72.9 ± 3.4	83.3 ± 4.6	75.7 ± 5.0	**87.3 ± 4.3**	**77.2 ± 3.8**
Energy (kcal)	2122 ± 663	1121 ± 189	**2266 ± 646**	**1317 ± 238**	**2432 ± 745**	**1541 ± 412**	**2375 ± 777**	**1531 ± 460**
Carbohydrate (g)	270.2 ± 58.3	168.3 ± 33.6	284.2 ± 75.4	167.3 ± 23.9	**318.0 ± 106.1**	**222.0 ± 97.5**	**313.6 ± 99.8**	**216.2 ± 85.7**
Protein (g)	83.4 ± 16.8	49.2 ± 6.3	**94.2 ± 26.6**	**55.5 ± 10.7**	**98.3 ± 29.5**	**62.3 ± 21.7**	**96.3 ± 29.5**	**62.5 ± 21.7**
Fat (g)	77.9 ± 23.8	44.5 ± 14.9	86.0 ± 30.6	49.7 ± 14.3	**88.7 ± 31.5**	**59.8 ± 21.6**	**84.4 ± 35.1**	**63.4 ± 31.4**
Starch (g)	129.9 ± 44.2	48.4 ± 15.5	132.8 ± 49.0	57.3 ± 17.1	**148.0 ± 58.1**	**77.6 ± 27.6**	**158.0 ± 61.1**	**81.7 ± 26.2**
Soluble Fibre (g)	2.6 ± 1.2	1.0 ± 0.4	2.6 ± 1.2	1.1 ± 0.4	2.8 ± 1.4	1.1 ± 0.4	3.0 ± 1.6	1.5 ± 0.6
Insoluble Fibre (g)	7.5 ± 3.4	3.2 ± 1.4	6.8 ± 3.3	3.5 ± 1.3	7.6 ± 4.3	3.6 ± 1.5	8.5 ± 6.6	4.8 ± 2.9
Glycaemic Index	55.3 ± 5.1	58.9 ± 7.3	54.5 ± 3.8	55.5 ± 4.3	55.9 ± 3.7	56.7 ± 5.7	**54.4 ± 3.4**	**56.5 ± 2.6**
Glycaemic Load	89.9 ± 27.4	47.7 ± 24.8	94.1 ± 24.1	48.8 ± 12.5	**110.1 ± 33.0**	**54.2 ± 14.8**	**113.1 ± 34.8**	**59.9 ± 16.7**
Saturated Fats (g)	23.3 ± 10.9	11.7 ± 4.1	**25.1 ± 9.6**	**14.9 ± 4.9**	**25.9 ± 9.4**	**18.4 ± 7.2**	**24.3 ± 9.7**	**20.8 ± 13.4**
Monounsaturated Fats (g)	27.5 ± 12.1	10.6 ± 6.4	**28.5 ± 12.3**	**14.1 ± 7.4**	**29.3 ± 11.2**	**16.5 ± 8.7**	**28.8 ± 13.3**	**16.7 ± 11.9**
Polyunsaturated Fats (g)	8.4 ± 3.4	3.8 ± 1.1	9.0 ± 3.1	4.6 ± 1.6	**9.6 ± 2.8**	**5.8 ± 2.8**	**10.0 ± 4.3**	**6.2 ± 3.2**
Omega-3 (% kcal/kcal tot)	0.4 ± 0.2	0.5 ± 0.2	0.4 ± 0.1	0.5 ± 0.2	0.4 ± 0.1	0.5 ± 0.2	0.4 ± 0.1	0.5 ± 0.4
Omega-6 (% kcal/kcal tot)	3.0 ± 1.1	2.5 ± 1.0	3.0 ± 0.6	2.7 ± 0.8	3.1 ± 0.4	2.7 ± 1.2	3.1 ± 0.6	2.8 ± 1.0
Cholesterol (mg)	238.9 ± 73.8	116.4 ± 58.5	**217.1 ± 73.5**	**95.0 ± 22.9**	265.1 ± 65.1	88.6 ± 60.3	236.5 ± 66.9	86.6 ± 60.5
Iron (mg)	9.0 ± 2.9	4.2 ± 1.0	8.8 ± 2.6	4.9 ± 1.7	9.4 ± 2.9	5.8 ± 2.8	**9.8 ± 3.9**	**6.3 ± 3.8**
Vitamin C (mg)	68.1 ± 30.4	39.0 ± 21.8	60.3 ± 22.1	37.5 ± 18.4	66.2 ± 26.6	45.5 ± 22.0	80.7 ± 49.8	59.3 ± 35.4
Vitamin A (mcg)	1081.0 ± 537.9	512.7 ± 604.1	982.2 ± 433.4	590.7 ± 469.6	980.4 ± 458.1	666.5 ± 623.9	910.3 ± 507.6	540.6 ± 328.9
Vitamin E (mg)	8.4 ± 3.8	3.6 ± 2.5	8.3 ± 3.8	4.5 ± 2.7	8.8 ± 4.4	5.2 ± 2.8	8.7 ± 4.7	5.2 ± 3.9

**Table 2 nutrients-12-00843-t002:** Factor Analysis results.

Component	Factor 1	Factor 2	Factor 3
Δ Fat	**0.865**	0.278	0.043
Δ Protein	**0.821**	0.088	0.395
Δ Carbohydrate	**0.731**	−0.030	0.466
Δ Energy	**0.713**	0.037	0.517
Δ Monounsaturated fat	**0.694**	0.326	0.206
Δ Saturated fat	**0.647**	0.003	0.209
Δ Vitamin E	**0.629**	0.439	0.392
Δ Omega 6	0.140	**0.848**	0.206
Δ Omega 3	0.016	**0.714**	0.061
Δ Polyunsaturated fat	0.472	**0.697**	0.444
Δ Insoluble fibre	0.282	0.281	**0.857**
Δ Soluble fibre	0.135	0.286	**0.836**
Δ Vitamin C	0.275	0.043	**0.783**
Δ Vitamin A	0.268	0.161	**0.644**
Δ Starch	0.551	0.118	**0.583**
Δ Iron	0.515	0.168	**0.573**
Eigenvalues	8.62	1.55	1.29
% of total variance	30.10	44.68	71.61

The underlined bold values indicate the components related to Factor 1, Factor 2 and Factor 3, respectively.

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
