# Peer review of "High-intensity Interval Training Promotes the Shift to a Health-Supporting Dietary Pattern in Young Adults"

_nutrients, 2020, doi:10.3390/nu12030843_

Round 1

Reviewer 1 Report

Dear Editor
I am glad to review the manuscript ID: nutrients-746101, entitled "High-intensity interval training promotes the shift to a healthier eating pattern in sedentary young subjects ". The authors addressed the question of whether high –intensity interval training (HIT) could affect “ spontaneous” food choice and designed an experiment, which enrolled 32 young adults (20 men and 2 women, normal BMI, age around 21-24 year-old). Their spontaneous diet behavior were monitored over nine-week of HIT (indoor-cycling training). The results showed this HIT program promotes a spontaneous modulation of food choices and regulation of dietary intake in these young adults.

In overall, the entire manuscript is well-written. The information delivered in this manuscript is quite important to preventive medicine - correct lifestyle now, secure health in the future. There are some points that remain further clarification in this study.

Title:

  1. “Healthier” eating pattern may not be precise. Usually it is not easy to tell between health, healthier and unhealthy due to much subject opinion.
  2. Also, it is better to change “young subjects” to “young adults” under consideration of respect to participants

Methods:

  1. Line 98 ketogentic or paleolithic diet, or others. Because of exclusion criteria, it has to state clearly. What does “others” refer to?
  2. Line 98-100 The definition of sedentary young adults are defined. It used a specific questionnaire that came from reference 14. However, this reference did not show either any specific questionnaire or definition of “ sedentary pattern”. Further, does it differ the current definition used in this study in varied ethnics?
  3. Did all participants complete this HIT program? Since these young adults were not regularly involved in exercise before this study, I really concern how they were able to complete it throughout this training course? Did any assistants help them during this period? if yes, how could avoid the assistant's effect? If no, it is better state clearly how these participants complete this HIT program?

Discussion:

A little more addressed in the findings between the HIT in this study and traditional constant low-to-moderate intensity exercise training course in young adults is suggested.

Author Response

Reviewer 1

We thank the Editor and the reviewers for their interesting and useful comments that helped us to enhance the quality of our paper. We do hope that thanks to the reviewers’ comments, we could successfully deal with the requested revisions, rendering the paper better readable. To facilitate the identification of the revisions, the “Track Changes” in Microsoft Word has been used for all new or modified sentences, according to the reviewers. 

Title:

Reviewer (R): “Healthier” eating pattern may not be precise. Usually it is not easy to tell between health, healthier and unhealthy due to much subjective opinion.

Also, it is better to change “young subjects” to “young adults” under consideration of respect to participants

Answer (A): The phrase“Healthier eating pattern” has been changed with “a health-supporting dietary pattern”. The term “subjects” has been replaced with “adults”, as suggested by the referee.

Methods:

R: Line 98 ketogenic or paleolithic diet, or others. Because of exclusion criteria, it has to state clearly. What does “others” refer to?

A: We revised the sentence, better explaining the term “others” as suggested by the referee. 

R: Line 98-100 The definition of sedentary young adults are defined. It used a specific questionnaire that came from reference 14. However, this reference did not show either any specific questionnaire or definition of “ sedentary pattern”. Further, does it differ the current definition used in this study in varied ethnics?

A: According to the reviewer's comment, we recognize that the set of questions taken from Malek and colleagues, doesn't allow to define "sedentary" a subject. However, our aim using that set of questions was to check the physical activity habits of the subjects (in terms of frequency, duration and intensity). This paragraph has been extended with more details and clarity has been improved. Furthermore, the VO2max values of the subjects directly measured during the incremental ramp test have been reported. In regards to the last reviewer’s question, the participants in our study were all Caucasians. 

R: Did all participants complete this HIT program? Since these young adults were not regularly involved in exercise before this study, I really concern how they were able to complete it throughout this training course? Did any assistants help them during this period? if yes, how could avoid the assistant's effect? If no, it is better state clearly how these participants complete this HIT program?

A: All the participants completed the training period. As now reported in the text “All participants completed the whole training program; three subjects had to quit a training session before the end due to personal issues, but this was not considered a problem for the statistical analyses.” As shown in Figure 1, the training period had a gradual incremental pattern (both in frequency and duration of each session) in order to allow every subject to complete the program. As reported by several authors in the literature, HIIT can be adapted to several types of populations, including sedentary one, since the relative intensity of the exercise is individualized. In fact, in our study, the training intensity was individualized based on the individual heart rate training zones of each participant.

All the training sessions were conducted by the same instructor, so an assistant's effect is unlikely.

Discussion:

R: A little more addressed in the findings between the HIT in this study and traditional constant low-to-moderate intensity exercise training course in young adults is suggested.

A: Some references about the differences between HIIT and MICT have been added in the discussion section. However, to widely discuss the differences between these two training modalities is not among the aims of this paper; this mainly because - as reported in the text - the effect of different training modes on dietary habits is scarce.

Reviewer 2 Report

To:

Editorial Board

Nutrients

Title: “High-intensity interval training promotes the shift to a healthier eating pattern in sedentary young subjects”

Dear Editor,

I read this manuscript and I think that:

  • The sample size is a limitation of the study design. This is a limitation and should be discussed in a dedicated limitation section.
  • A post-hoc sample size calculation should be provided.
  • Results section of the abstract should be implemented by including more numerical data. Please provide.
  • A multivariate regression analysis should be provided to evaluate the role of confounding factors on final results.
  • the role of nutraceuticals should also be considered in healthy diet. Please discuss such a point in a dedicated limitation section. Please provide.
